# FULLY STEERABLE 3D SPHERICAL NEURONS

## ABSTRACT

Emerging from low-level vision theory, steerable filters found their counterpart in prior work on steerable convolutional neural networks equivariant to rigid transformations. In our work, we propose a steerable feed-forward learning-based approach that consists of spherical decision surfaces and operates on point clouds. Focusing on 3D geometry, we derive a 3D steerability constraint for hypersphere neurons, which are obtained by conformal embedding of Euclidean space and have recently been revisited in the context of learning representations of point sets. Exploiting the rotational equivariance, we show how our model parameters are fully steerable at inference time. We use a synthetic point set and real-world 3D skeleton data to show how the proposed spherical filter banks enable making equivariant and, after online optimization, invariant class predictions for known point sets in unknown orientations.

## 1 INTRODUCTION

We present a novel feed-forward model consisting of steerable 3D neurons for point cloud classification, an important and challenging problem with many applications such as autonomous driving, human-robot interaction, and mixed-reality installations. We achieve the steerability by using geometric neurons (Melnyk et al., 2021), leading to a rotation equivariant network. Besides becoming a geometrically explainable approach, further benefits of the proposed method include rotation invariant classification and, therefore, lowered data augmentation requirements during learning.

We make use of a conformal embedding to obtain higher-order decision surfaces. Following the motivation in the recent work of Melnyk et al. (2021), we focus on 3D geometry and spherical decision surfaces, arguing for their natural suitability for problems in Euclidean space. We show how a *spherical neuron (classifier)*, i.e., the hypersphere neuron (Banarer et al., 2003b) or its generalization for 3D input point sets — the geometric neuron (Melnyk et al., 2021) — can be turned into a steerable neuron. We prove that the aforementioned spherical neurons in any dimension require only up to first-degree spherical harmonics to accommodate the effect of rotation. This allows us to derive a 3D steerability constraint for the spherical neurons and to describe a recipe for creating a steerable model from a pretrained spherical classifier.

Using the synthetic Tetris dataset (Thomas et al., 2018) and the skeleton data from the UTKinect-Action3D dataset (Xia et al., 2012), we first verify the derived constraint and check its stability with respect to perturbations in the input. We further conduct an experiment in a realistic setting, where we initialize the steerable model parameters using a noisy or imperfect prediction of the transformation applied to the input and optimize the parameters in an unsupervised way.

The contributions of our work are as follows:
(**a**) We prove that the activation of spherical neurons on rotated input only varies by up to first-degree spherical harmonics (Section 4.1.1).
(**b**) Based on a minimal set of four spherical neurons that are rotated to the corresponding vertices of a regular tetrahedron (Section 4.1.2), we derive the main result of our paper (Section 4.1.3) – a 3D steerability constraint for spherical neurons (13).
(**c**) We propose a method to turn a trained spherical classifier into a fully steerable model (Section 5.2) that produces predictions invariant to 3D rotations, even in the presence of noise (Section 5.3), which we verify experimentally using both synthetic and real 3D data.

## 2 RELATED WORK

### 2.1 STEERABILITY AND EQUIVARIANCE

Steerability is a powerful concept from early vision and image processing (Freeman et al., 1991; Knutsson et al., 1992; Simoncelli et al., 1992; Perona, 1995; Simoncelli & Freeman, 1995; Teo & Hel-Or, 1998) that resonates in the era of deep learning. The utility of steerable filters and the main theorems for their construction are presented in the seminal work of Freeman et al. (1991). Geometric equivariance in the context of computer vision has been an object of extensive research over the last decades (Van Gool et al., 1995). Equivariance is a necessary property for steerability as the group acting on the input space needs to be represented in the co-domain of the operator. For this reason, Lie theory can be used to to study steerable filters for equivariance and invariance (Reisert, 2008).

Nowadays, equivariance-related research extends into deep learning, e.g., the SE(3)-equivariant models of Fuchs et al. (2020), Thomas et al. (2018), and Zhao et al. (2020), and the SO(3)-equivariant network of Anderson et al. (2019), as well as Marcos et al. (2017) and Kondor et al. (2018). Also steerable filter concepts are increasingly used in current works, e.g., Cohen & Welling (2016) considered image data and proposed a CNN architecture to produce equivariant representations with steerable features, which involves fewer parameters than traditional CNNs. They considered the dihedral group (*discrete* rotation, translation, and reflection), and the steerable representations in their work are proposed as formation of elementary feature types. One limitation of their approach is that rotations are restricted to four orientations, i.e., by $\pi/2$. More recently, Weiler et al. (2018b) utilized group convolutions and introduced steerable filter convolutional neural networks (SFCNNs) operating on images to jointly attain equivariance under translations and discrete rotations. In their work, the filter banks are learned rather than fixed. Further, the work of Weiler & Cesa (2019) proposed a unified framework for E(2)-equivariant steerable CNNs and presented their general theory.

The steerable CNNs for 3D data proposed by Weiler et al. (2018a) are closely related to our work. The authors employed a combination of scalar, vector and tensor fields as features transformed by SO(3) representations and presented a model that is equivariant to SE(3) transformations. They also considered different types of nonlinearities suitable for nonscalar components of the feature space. The novel SE(3)-equivariant approach by Fuchs et al. (2020) introduced a self-attention mechanism that is invariant to global rotations and translations of its input and solves the limitation of angularly constrained filters in other equivariant methods, e.g., Thomas et al. (2018). Noteworthy, the work of Jing et al. (2020) proposed the geometric vector perceptron (GVP) consisting of two linear transformations for the input scalar and vector features, followed by nonlinearities. The GVP scalar and vector outputs are invariant and equivariant, respectively, with respect to an arbitrary composition of rotations and reflections in the 3D Euclidean space.

The key point distinguishing our approach from other equivariant networks is that we use conformal modeling (Li et al., 2001a; Hitzer, 2008), with which the scalar product of two points yields their Euclidean distance (allowing us to construct spherical decision surfaces) and nonlinear conformal transformations on $\mathbb{R}^n$ can be linearized (allowing us to steer the decision spheres). Besides, we do not constrain the space of learnable parameters (as opposed to, e.g., Thomas et al. (2018), Weiler et al. (2018a), and Fuchs et al. (2020)), but construct our steerable model from a freely trained base network, as we discuss in detail in Section 4.

### 2.2 CONFORMAL MODELING AND THE HYPERSPHERE NEURON

The utility of conformal embedding for Euclidean geometry and the close connection to Minkowski spaces are thoroughly discussed by Li et al. (2001a). An important result is that one can construct hyperspherical decision surfaces using representations in the conformal space (Li et al., 2001b), as done in the work of Perwass et al. (2003). The hypersphere neuron proposed by Banarer et al. (2003b) is such a spherical classifier. Remarkably, since a hypersphere can be seen as a generalization of a hyperplane, the standard neuron can be considered as a special case of the hypersphere neuron. Stacking multiple hypersphere neurons in a feed-forward network results in a multilayer hypersphere perceptron (MLHP), which was shown by Banarer et al. (2003a) to outperform the standard MLP for some classification tasks. However, its application to point sets was not discussed. This motivated the work of Melnyk et al. (2021) on the geometric neuron, where the learned parameters were shown to represent a combination of spherical decision surfaces. Moreover, the geometric neuron activations

were proved to be isometric in the 3D Euclidean space. In our work, we use the latter observation as the necessary condition for deriving the steerability constraint. Making spherical classifiers *steerable* adds to the practical value of the prior work by Melnyk et al. (2021).

## 3 BACKGROUND

### 3.1 STEERABILITY

As per Freeman et al. (1991), a 2D function $f(x, y)$ is said to steer if it can be written as a linear combination of rotated versions of itself, i.e., when it satisfies the constraint

$$f^\theta(x, y) = \sum_{j=1}^{M} v_j(\theta) f^{\theta_j}(x, y) , \tag{1}$$

where $v_j(\theta)$ are the interpolation functions, $\theta_j$ are the basis function orientations, and $M$ is the number of basis function terms required to steer the function. An alternative formulation can be found in the work of Knutsson et al. (1992). In 3D, the steering equation becomes

$$f^R(x, y, z) = \sum_{j=1}^{M} v_j(R) f^{R_j}(x, y, z) , \tag{2}$$

where $f^R(x, y, z)$ is $f(x, y, z)$ rotated by $R \in \mathrm{SO}(3)$, and each $R_j \in \mathrm{SO}(3)$ orients the corresponding $j$th basis function.

Theorems 1, 2 and 4 in Freeman et al. (1991) describe the conditions under which the steerability constraints (1) and (2) hold, and how to determine the minimum number of basis functions for the 2D and 3D case, respectively.

### 3.2 CONFORMAL EMBEDDING

We refer the reader to Section 3 in the work of Melnyk et al. (2021) for more details, and only briefly introduce important notation in this section. The conformal space for the Euclidean $\mathbb{R}^n$ counterpart can be formed as $\mathbb{ME}^n \equiv \mathbb{R}^{n+1,1} = \mathbb{R}^n \oplus \mathbb{R}^{1,1}$, where $\mathbb{R}^{1,1}$ is the Minkowski plane (Li et al., 2001a) with orthonormal basis defined as $\{e_+, e_-\}$ and null basis $\{e_0, e_\infty\}$ representing the origin $e_0 = \frac{1}{2}(e_- - e_+)$ and point at infinity $e_\infty = e_- + e_+$. Thus, a Euclidean vector $\mathbf{x} \in \mathbb{R}^n$ can be embedded in the conformal space $\mathbb{ME}^n$ as

$$X = \mathcal{C}(\mathbf{x}) = \mathbf{x} + \frac{1}{2}\|\mathbf{x}\|^2 e_\infty + e_0 , \tag{3}$$

where $X \in \mathbb{ME}^n$ is called *normalized*. The conformal embedding (3) represents the stereographic projection of $\mathbf{x}$ onto a projection sphere in $\mathbb{ME}^n$ and is *homogeneous*, i.e., all embedding vectors in the equivalence class $[X] = \{Z \in \mathbb{R}^{n+1,1} : Z = \gamma X, \ \gamma \in \mathbb{R} \setminus \{0\}\}$ are taken to represent the same vector $\mathbf{x}$. Importantly, given the conformal embedding $X$ and some $Y = \mathbf{y} + \frac{1}{2}\|\mathbf{y}\|^2 e_\infty + e_0$, their scalar product in the conformal space corresponds to their Euclidean distance, $X \cdot Y = -\frac{1}{2}\|\mathbf{x} - \mathbf{y}\|^2$. This interpretation of the scalar product in the conformal space is the main motivation in constructing spherical classifiers.

### 3.3 SPHERICAL CLASSIFIERS

By *spherical classifiers*, we collectively refer to hypersphere (Banarer et al., 2003a) and geometric (Melnyk et al., 2021) neurons, which have spherical decision surfaces.

As discussed by Banarer et al. (2003a), by embedding both a data vector $\mathbf{x} \in \mathbb{R}^n$ and a hypersphere $S \in \mathbb{ME}^n$ in $\mathbb{R}^{n+2}$ as

$$\boldsymbol{X} = \left(x_1, \ldots, x_n, -1, -\frac{1}{2}\|\mathbf{x}\|^2\right) \in \mathbb{R}^{n+2}, \qquad \boldsymbol{S} = \left(c_1, \ldots, c_n, \frac{1}{2}(\|\mathbf{c}\|^2 - r^2), 1\right) \in \mathbb{R}^{n+2} , \tag{4}$$

where $\mathbf{c} = (c_1, \ldots, c_n) \in \mathbb{R}^n$ is the hypersphere center and $r \in \mathbb{R}$ is the radius, their scalar product $X \cdot S$ in the conformal space $\mathbb{ME}^n$ can be computed equivalently in $\mathbb{R}^{n+2}$ as $\boldsymbol{X}^\top \boldsymbol{S}$:

$$X \cdot S = \boldsymbol{X}^\top \boldsymbol{S} = -\frac{1}{2}\|\mathbf{x} - \mathbf{c}\|^2 + \frac{1}{2}r^2 . \tag{5}$$

This result enables the implementation of a hypersphere neuron in $\mathbb{ME}^n$ using the standard dot product in $\mathbb{R}^{n+2}$. The hypersphere vector components are treated as independent learnable parameters during training. Thus, a spherical classifier effectively learns *non-normalized* hyperspheres of the form $\widetilde{S} = (s_1, \ldots, s_{n+2}) \in \mathbb{R}^{n+2}$. Due to the homogeneity of the representation, both normalized and non-normalized hyperspheres represent the same decision surface. More details can be found in Section 3.2 in the work of Melnyk et al. (2021).

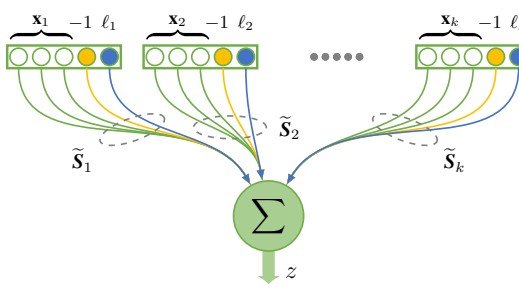

Figure 1: The geometric neuron ($\ell_k = -\frac{1}{2}\|\mathbf{x}_k\|^2$).

The geometric neuron is a generalization of the hypersphere neuron for points sets as input, see Figure 1. A single geometric neuron output is thus the sum of the signed distances of $k$ input points to $k$ learned hyperspheres

$$z = \sum_{k=1}^{N} \gamma_k \, X_k^\top S_k \,, \tag{6}$$

where $z \in \mathbb{R}$, $X_k \in \mathbb{R}^5$ is a properly embedded 3D input point, $\gamma_k \in \mathbb{R}$ is the scale factor, i.e., the last element of the learned parameter vector $\widetilde{S}_k$, and $S_k = \widetilde{S}_k / \gamma_k \in \mathbb{R}^5$ are the corresponding normalized learned parameters (spheres).

Furthermore, Melnyk et al. (2021) demonstrated that the geometric neuron activations are isometric in 3D. That is, rotating the input is equivalent to rotating the decision spheres. This result is a necessary condition to consider rotation and translation equivariance of models constructed with geometric neurons and forms the basis for our methodology.

In the following sections, we use the same notation for a 3D rotation $R$ represented in the Euclidean space $\mathbb{R}^3$, the homogeneous (projective) space $P(\mathbb{R}^3)$, and $\mathbb{ME}^3 \cong \mathbb{R}^5$, depending on the context. This is possible since we can add the required number of ones to the diagonal of the original rotation matrix without changing the transformation representation.

## 4 METHOD

To build a 3D steerable model, we perform the following steps: We first train an MLGP model (Melnyk et al., 2021), which consists of spherical neurons. After optimizing the model parameters, we freeze them and transform according to the 3D steerability constraint we derive. Finally, by combining the resulting parameters in filter banks and adding interpolation coefficients as free parameters, we create a steerable model.

### 4.1 FULLY STEERABLE 3D SPHERICAL NEURONS

In this section, we identify the conditions under which a geometric neuron as a function of its 3D input can be steered. In other words, we derive an expression that gives us the response of a *hypothetical* geometric neuron for some input, using rotated versions of the learned geometric neuron parameters. We start by considering the steerability conditions for a single sphere classifying the corresponding input point $X$, i.e., $f(X) = X^\top S$, where $X$ and $S$ are embedded in $\mathbb{ME}^3 \cong \mathbb{R}^5$ according to (4). Since the geometric neuron output is a linear combination of these functions, as per (6), we will use the identified conditions to build a steerable feed-forward network by reassembling modified spheres.

#### 4.1.1 BASIS CONSTRUCTION

To formulate a steerability constraint for a spherical neuron (sphere), first, we need to determine the minimum number of basis functions, i.e., the number of terms $M$ in (2). This number only depends on the degree of the spherical harmonics that are required to compute the steered result (Freeman et al., 1991). Thus, we need to determine the required degrees.

**Theorem.** *Let $S \in \mathbb{R}^{n+2}$ be an $n$D classifier with center $c \in \mathbb{R}^n$ ($R := \|c\|$) and radius $r$, and $x \in \mathbb{R}^n$ be a point represented by $X \in \mathbb{R}^{n+2}$, see (4). Let further $S'$ be the classifier that is obtained by rotating $S$ in $n$D space, then $X^\top S'$ and $X^\top S$ are related by spherical harmonics up to first degree.*

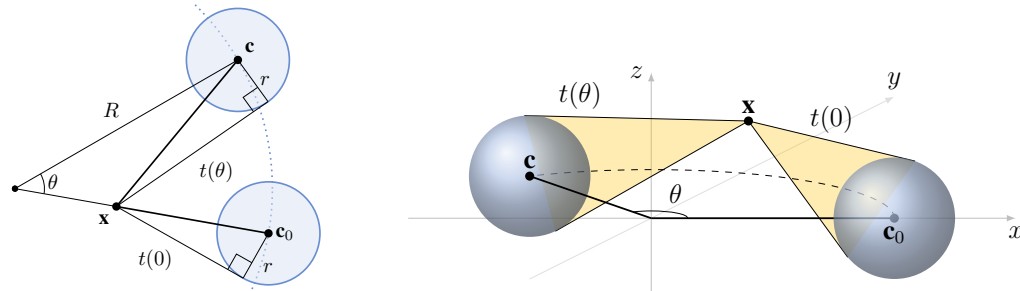

Figure 2: The effect of rotation on the spherical neuron activation in 2D (left) and 3D (right); $t(\theta)$ denotes the tangent length.

*Proof.* Without loss of generality, the rotation is defined by the plane of rotation $\boldsymbol{\pi}$ by the angle $\theta$. Denote the projection of a vector $\mathbf{v} \in \mathbb{R}^n$ onto $\boldsymbol{\pi}$ by $\mathbf{v}_{\boldsymbol{\pi}}$ and define $\mathbf{v}_{\perp\boldsymbol{\pi}} = \mathbf{v} - \mathbf{v}_{\boldsymbol{\pi}}$. From (5) we obtain

$$2\boldsymbol{X}^\top \boldsymbol{S} = r^2 - \|\mathbf{x} - \mathbf{c}\|^2 = r^2 - \left\|(\mathbf{x} - \mathbf{c})_{\perp\boldsymbol{\pi}}\right\|^2 - \left\|(\mathbf{x} - \mathbf{c})_{\boldsymbol{\pi}}\right\|^2 \quad .$$

A rotation in $\boldsymbol{\pi}$ only affects the rightmost term above and there exists a $\phi \in [0, 2\pi)$ such that

$$\left\|(\mathbf{x} - \mathbf{c})_{\boldsymbol{\pi}}\right\|^2 = \|\mathbf{x}_{\boldsymbol{\pi}} - \mathbf{c}_{\boldsymbol{\pi}}\|^2 = \|\mathbf{x}_{\boldsymbol{\pi}}\|^2 + \|\mathbf{c}_{\boldsymbol{\pi}}\|^2 - 2\|\mathbf{x}_{\boldsymbol{\pi}}\| \|\mathbf{c}_{\boldsymbol{\pi}}\| \cos \phi \quad .$$

With a similar argument, we obtain

$$2\boldsymbol{X}^\top \boldsymbol{S}' = r^2 - \left\|(\mathbf{x} - \mathbf{c})_{\perp\boldsymbol{\pi}}\right\|^2 - \|\mathbf{x}_{\boldsymbol{\pi}}\|^2 - \|\mathbf{c}_{\boldsymbol{\pi}}\|^2 + 2\|\mathbf{x}_{\boldsymbol{\pi}}\| \|\mathbf{c}_{\boldsymbol{\pi}}\| \cos(\phi + \theta) \quad . \qquad \square$$

This result is valid in any dimension, but we are primarily interested in $n = 2$ and $n = 3$, as illustrated in Figure 2. Following the result of Theorem 4 in Freeman et al. (1991) and using $N = 1$, we have that $M = (N + 1)^2 = 4$ basis functions suffice in the 3D case (2).

### 4.1.2 SPHERICAL FILTER BANKS IN 3D

In 3D, we thus select four rotated versions of the function $f(\boldsymbol{X})$, as the basis functions. The rotations $\{\boldsymbol{R}_j\}_{j=1}^4$ must be chosen to satisfy the condition (b) in Theorem 4 (Freeman et al., 1991). Therefore, we transform $f(\boldsymbol{X})$ such that the resulting four spheres are spaced in three dimensions equally, i.e., form a regular tetrahedron with the vertices $(1, 1, 1)$, $(1, -1, -1)$, $(-1, 1, -1)$, and $(-1, -1, 1)$, as shown in Figure 3 b). We stack the homogeneous coordinates of the tetrahedron vertices $\mathbf{m}_j$ in a matrix column-wise (scaled by $1/2$) to get the orthogonal matrix

$$\mathbf{M} = \begin{bmatrix} \mathbf{m}_1 & \mathbf{m}_2 & \mathbf{m}_3 & \mathbf{m}_4 \end{bmatrix} = \frac{1}{2} \begin{bmatrix} 1 & 1 & -1 & -1 \\ 1 & -1 & 1 & -1 \\ 1 & -1 & -1 & 1 \\ 1 & 1 & 1 & 1 \end{bmatrix} . \tag{7}$$

We will use this matrix operator $\mathbf{M}$ to compute the linear coefficients in the vector space generated by the vertices of the regular tetrahedron (Granlund & Knutsson, 1995). This will be necessary to find the appropriate interpolation functions and formulate the steerability constraint in Section 4.1.3.

The four rotated versions of the function $f(\boldsymbol{X})$ will constitute the basis functions that we call a spherical filter bank. To construct this filter bank, we choose the following convention. The originally learned spherical classifier $f(\boldsymbol{X}) = \boldsymbol{X}^\top \boldsymbol{S}$ is first rotated to $(1, 1, 1)$ (see Figure 3 b)) with the corresponding transformation denoted as $\boldsymbol{R}_O$. Next, we rotate the transformed sphere into the other three vertices of the regular tetrahedron and transform back to the original coordinate system (see Figure 3 a) for the case of $(1, -1, -1)$). The resulting filter bank for one spherical classifier is thus composed as the following $20 \times 1$ matrix:

$$B(\boldsymbol{S}) = \left[ \boldsymbol{R}_O^\top \boldsymbol{R}_{Ti} \boldsymbol{R}_O \boldsymbol{S} \right] , \tag{8}$$

where each of $\{\boldsymbol{R}_{Ti}\}_{i=0}^3$ is the rotation isomorphism in $\mathbb{R}^5$ corresponding to a 3D rotation from $(1, 1, 1)$ to the vertex $i + 1$ of the regular tetrahedron. Therefore, $\boldsymbol{R}_{T0} = \mathbf{I}_5$.

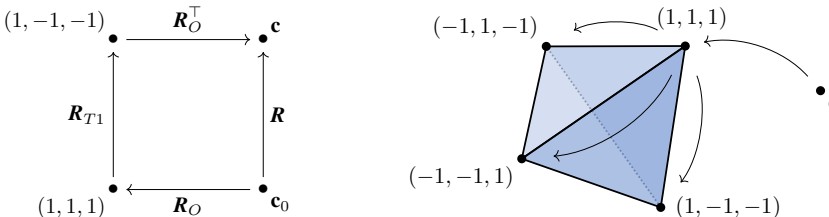

Figure 3: a) A rotation from $\mathbf{c}_0$ to $\mathbf{c}$ described by a tetrahedron rotation. b) A regular tetrahedron.

### 4.1.3 3D STEERABILITY CONSTRAINT

The steerability constraint can be formulated as follows. For an arbitrary orientation $\boldsymbol{R}$ applied to the input of the function $f(\boldsymbol{X})$, we want the output of the spherical filter bank $B(\boldsymbol{S})$ in (8) to be interpolated with $v_j(\boldsymbol{R})$ such that the response is equal to the original function output, i.e.,

$$f(\boldsymbol{X}) = f^{\boldsymbol{R}}(\boldsymbol{R}\boldsymbol{X}) = \sum_{j=1}^{M} v_j(\boldsymbol{R})\, f^{\boldsymbol{R}_j}(\boldsymbol{R}\boldsymbol{X}) = (v(\boldsymbol{R}) \otimes \boldsymbol{R}\boldsymbol{X})^{\top} B(\boldsymbol{S}), \tag{9}$$

where $\boldsymbol{X} \in \mathbb{R}^5$ is a single, appropriately embedded, 3D point, $v(\boldsymbol{R}) \in \mathbb{R}^4$ is a vector of the interpolation coefficients, and $\otimes$ denotes the Kronecker product (Horn & Johnson, 1991).

The coefficients $v(\boldsymbol{R})$ should conform to the basis function construction (condition (b) in Theorem 4 in Freeman et al. (1991)), which is why they are computed with $\mathbf{M}$ defined in (7). Given $\boldsymbol{X} \in \mathbb{R}^5$ as input and an unknown rotation $\boldsymbol{R}$ acting on it, the steering equation (9) implies

$$(v(\boldsymbol{R}) \otimes \boldsymbol{R}\boldsymbol{X})^{\top} B(\boldsymbol{S}) = (v(\mathbf{I}) \otimes \boldsymbol{X})^{\top} B(\boldsymbol{S}) . \tag{10}$$

Given a tetrahedron rotation, e.g., $\boldsymbol{R}_{T1}$, as shown in the diagram in Figure 3 a), we can define the unknown rotation accordingly as $\boldsymbol{R} = \boldsymbol{R}_O^{\top} \boldsymbol{R}_{T1} \boldsymbol{R}_O$. In this case, it is easy to see that to satisfy the constraint (10), $v(\boldsymbol{R})$ must be $(0, 1, 0, 0)$, i.e., the second filter in the filter bank $B(\boldsymbol{S})$ must be chosen. This can be achieved by transforming a constant vector $\mathbf{m}_1 = (\frac{1}{2}, \frac{1}{2}, \frac{1}{2}, \frac{1}{2})$ by rotation $\boldsymbol{R}_{T1}$ and multiplying it by the basis matrix $\mathbf{M}$ as follows:

$$v(\boldsymbol{R}) = \mathbf{M}^{\top} (\boldsymbol{R}_{T1}\, \mathbf{m}_1) = \mathbf{M}^{\top} \left(\frac{1}{2}, -\frac{1}{2}, -\frac{1}{2}, \frac{1}{2}\right) = (0, 1, 0, 0) . \tag{11}$$

Note that, in general, a geometric neuron (6) takes a *set* of embedded points as input. Therefore, with the setup above, $\boldsymbol{R}$ will be different for each input shape point $k$ if the same $v$ is used for all $k$, which contradicts that the shape is transformed by a rigid body motion, i.e., the same $\boldsymbol{R}_B$ for all $k$. Thus, we need to consider a suitable vector $v^k$ for each input shape point $k$, such that the resulting $\boldsymbol{R}_B$ is the same for all $k$. This can be achieved recalling how we construct the basis functions in the spherical filter bank (8): we need to consider the respective initial rotation $\boldsymbol{R}_O^k$. The desired interpolation coefficients $v^k$ are thus computed as

$$v^k(\boldsymbol{R}_B) = \mathbf{M}^{\top} (\boldsymbol{R}_O^k \boldsymbol{R}_B \boldsymbol{R}_O^{k\top} \mathbf{m}_1) . \tag{12}$$

The resulting $v^k(\boldsymbol{R}_B)$ interpolate the responses of the tetrahedron-copies of the originally learned sphere $\boldsymbol{S}_k$ to replace the rotated sphere.

By plugging (9) into (6), we can now establish the main result of our paper – the steerability constraint for a geometric neuron that takes a set of $N$ embedded points $\{\boldsymbol{X}_k\}_k$ as input:

$$f^{\boldsymbol{R}}(\boldsymbol{R}\boldsymbol{X}) = \sum_{k=1}^{N} \gamma_k\, f^{k\,\boldsymbol{R}}(\boldsymbol{R}\boldsymbol{X}_k) = \sum_{k=1}^{N} \gamma_k\, (v^k(\boldsymbol{R}) \otimes \boldsymbol{R}\boldsymbol{X}_k)^{\top} B(\boldsymbol{S}_k) . \tag{13}$$

## 5 EXPERIMENTS

We use a single GTX 1050 Ti GPU and conduct two types of experiments to confirm our findings presented in Section 4.

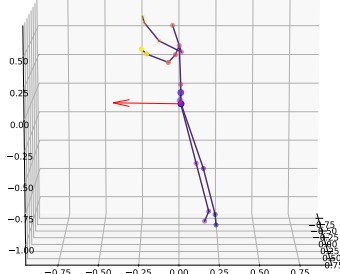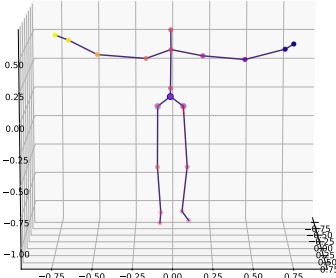

Figure 4: The effect of standardizing the orientation of the 3D skeleton representing *waveHands* action: the original (left) and the derotated (right) shape; the arrow is the normal vector of the plane formed by the three hip joints.

## 5.1 DATASETS

**3D Tetris**  Following the experiments reported by Thomas et al. (2018), Weiler et al. (2018a), and Melnyk et al. (2021), we use the synthetic point set of eight 3D Tetris shapes consisting of four points each (see, e.g., Figure 3 in Melnyk et al. (2021) and the numerical description in Appendix A).

**3D skeleton data**  We also perform experiments on real-world data to substantiate the validity of our theoretical results. We use the UTKinect-Action3D dataset introduced by Xia et al. (2012), in particular, the 3D skeletal joint locations extracted from Kinect depth maps. For each action sequence and from each frame, we extract the skeleton consisting of twenty points and assign it to the class of actions (ten categories) this frame is part of. Therefore, we formulate the task as shape recognition (i.e., a kind of action recognition from a static point cloud), where each shape is of size $20 \times 3$. Since the orientations of the shapes vary significantly across the sequences, we perform the following standardization: We first center each shape at the origin, and then compute the orientation from its three hip joints and derotate the shape in the $xy$-plane (viewer coordinate system) accordingly. We illustrate the effect of derotation in Figure 4. From each action sequence, we randomly select 50% of the skeletons for the test set and 20% of the remainder as validation data. The resulting data split is as follows: 2295 training shapes, 670 shapes for validation, and 3062 test shapes.

## 5.2 STEERABLE MODEL CONSTRUCTION

To construct and test steerable models, we perform the same steps for both datasets. The minor differences are the choice of training hyperparameters and the presence of validation and test subsets in the 3D skeleton dataset.

We first train a two-layer MLGP model (Melnyk et al., 2021), where the hidden layer consists of geometric neurons and the output layer of hypersphere neurons, to classify the shapes. Since the architecture choice is not the objective of the experiments, when building the MLGP, we use only one configuration with five hidden units for the Tetris data and twelve hidden units (determined by using the validation data) for the 3D skeleton dataset throughout the experiments. Similar to Melnyk et al. (2021), we do not use any activation function in the hidden layer due to the nonlinearity of the conformal embedding. We implement both MLGP models in PyTorch (Paszke et al., 2019) and keep the default parameter initialization for the linear layers. We train both models by minimizing the cross-entropy loss function and use the Adam optimizer (Kingma & Ba, 2015) with the default hyperparameters (the learning rate is 0.001). The Tetris MLGP learns to classify the eight shapes in the canonical orientation perfectly after 2000 epochs, whereas the Skeleton model trained for 10000 epochs achieves a test set accuracy of 92.9%. We refer to both trained models as *ancestor* MLGP.

For both, we then freeze the trained parameters and construct a steerable model. Note that we form steerable units only in the hidden layer and keep the output, i.e., classification, layer hypersphere neurons as they are. The steerability is not required for the subsequent layers as the output of the first layer becomes rotation-independent. The steerable units are formed from the corresponding frozen parameters as the (fixed) filter banks according to (8).

Table 1: Known rotation experiment: the steerable model classification accuracy for the distorted (the noise units are specified in the square brackets) rotated shapes (mean and std over 1000 runs, %).

| | 3D Tetris | | | 3D skeleton data (*test* set) | |
|---|---|---|---|---|---|
| Noise ($a$), [1] | **Steerable** | Ancestor | Noise ($a$), [m] | **Steerable** | Ancestor |
| 0.00 | **100.0 ± 0.0** | $47.3 \pm 34.0$ | 0.000 | **92.9 ± 0.0** | $25.2 \pm 23.1$ |
| 0.05 | **100.0 ± 0.0** | $47.6 \pm 34.2$ | 0.005 | **92.4 ± 0.2** | $24.6 \pm 22.1$ |
| 0.10 | **100.0 ± 0.0** | $49.3 \pm 35.0$ | 0.010 | **91.1 ± 0.3** | $24.4 \pm 20.7$ |
| 0.20 | **100.0 ± 0.4** | $46.0 \pm 34.4$ | 0.020 | **87.1 ± 0.5** | $23.5 \pm 20.6$ |
| 0.30 | **99.7 ± 1.9** | $47.6 \pm 34.4$ | 0.030 | **82.3 ± 0.6** | $24.3 \pm 20.3$ |
| 0.50 | **94.9 ± 7.7** | $44.5 \pm 31.9$ | 0.050 | **72.0 ± 0.7** | $22.8 \pm 17.5$ |

The only free parameters of this constructed steerable model are interpolation coefficients $v^k(\boldsymbol{R}_B) \in \mathbb{R}^4$ defined in (12), where $k$ indexes the learned hidden layer parameters (spheres) in the ancestor MLGP model. But since the only variables in (12) are the parameters of the unknown rotation $\boldsymbol{R}_B$ acting on the input, we can directly optimize them if they are unknown. We select the *axis–angle* rotation representation because it has only three parameters — the three coordinates of the rotation axis scaled by the rotation angle.

We propose the following basic recipe to create a steerable model: Train the ancestor MLGP $\rightarrow$ Fix the learned parameters $\rightarrow$ Transform the hidden unit parameters into filter banks (8) $\rightarrow$ Add the interpolation coefficients $v^k$ as free parameters to fulfill (13) $\rightarrow$ Steerable spherical classifier.

### 5.3 KNOWN ROTATION EXPERIMENT

Using the trained MLGP, we verify the correctness of (13). We first rotate the original data and then use this ground truth rotation to initialize the interpolation coefficients of the constructed steerable model according to (12). Our intuition is that if the steerability constraint (13) is correct, then, given the transformed point set, the activations of the steerable units in the steerable model will be equal to the activations of the geometric neurons in the ancestor MLGP model fed with the point set in the canonical orientation. Hence, the classification accuracies of the ancestor and steerable models on the original and transformed datasets, respectively, should be equal.

We run this experiment 1000 times. Each time, we generate a random rotation and apply it to the original point set (in case of the 3D skeleton data, we use the *test* split). We use these rotation parameters to create a steerable model and evaluate it on the transformed point set. To verify the stability of the steerable unit activations, we add uniform noise to the transformed points, $\boldsymbol{n} \sim U(-a, a)$, where the range of $a$ is motivated by the magnitude of the points in the datasets. For reference, we also present the accuracy of the ancestor model classifying the rotated data. We summarize the results in Table 1. The respective L1 distance results are presented in Appendix B. The analogous Tetris experiment results reported in related work are $100\%$ (over 25 runs and including random translation) (Thomas et al., 2018) and $99 \pm 2\%$ (over 17 runs) (Weiler et al., 2018a). However, due to the significant conceptual difference between our approach and these methods, as mentioned at the end of Section 2.1, the results are not directly comparable.

### 5.4 ESTIMATED ROTATION EXPERIMENT

In this experiment, we feed some rotation information into the system. For example, this information can come from another network branch that performs regression on the rotation, or from a dynamical model as in tracking. In either case, we cannot assume that the rotation is entirely accurate, unlike the experiment in Section 5.3. Therefore, we check the stability with respect to the noise in the rotation. Same as before, we exploit the MLGP trained on the respective point set in the canonical orientation, as described in Section 5.2, to build the steerable model. At each experiment run, we randomly select one shape (from the *test* subset in case of the 3D skeleton data) and apply a random rotation, $\boldsymbol{R}_{\mathrm{GT}}$, to it. We then perturb this ground truth rotation by varying the rotation angle. We achieve this by sampling the rotation angle from a Gaussian distribution with $\sigma = \pi/18 = 10°$ and various means, constructing a rotation of this sampled angle about a random axis, $\boldsymbol{R}_{\mathrm{noise}}$, and multiplying with the ground truth: $\boldsymbol{R}_{\mathrm{init}} = \boldsymbol{R}_{\mathrm{noise}} \boldsymbol{R}_{\mathrm{GT}}$. We use the resulting rotation $\boldsymbol{R}_{\mathrm{init}}$ in the axis-angle representation to initialize the steerable model parameters.

Table 2: Estimated rotation experiment: the steerable model classification accuracy (mean over 1000 runs, %) for a randomly chosen rotated shape before and after online optimization and different distortion rotation angle $\sim \mathcal{N}(\mu, \sigma = \pi/18)$.

| | 3D Tetris | | 3D skeleton data (*test* set) | |
|---|---|---|---|---|
| Noise angle mean ($\mu$) | Initial | Final | Initial | Final |
| 0 | 99.9 | 99.9 | 76.8 | 76.9 |
| $\pi/36$ | 99.8 | 99.9 | 80.0 | 80.3 |
| $\pi/18$ | 99.8 | 99.8 | 73.9 | 74.1 |
| $\pi/12$ | 99.2 | 99.3 | 68.8 | 68.9 |
| $\pi/6$ | 93.6 | 94.1 | 44.6 | 44.6 |

Further, we perform the optimization of the rotation vector parameters, which we call *online* optimization. Since inference time, we do not have access to the labels. Thus, we need to select a loss based on the consistency of the output. Here, we choose the entropy, i.e., $\mathcal{L}(\mathbf{p}) = -\sum_{i=1}^{K} p_i \log(p_i)$, where $\mathbf{p} \in \mathbb{R}^K$ is the softmax output of the model. Our intuition is that by minimizing the entropy in the output with respect to the steerable model parameters, we effectively force the model to produce a more confident prediction. For this optimization procedure, we use Adam with learning rate equal to 0.01, and empirically determine the number of epochs of 300 to be sufficient for the Tetris data and 100 for the 3D skeleton data.

Note that at each run, we provide the model with only one transformed shape, which implies that the accuracy per run is binary, i.e., the shape is either recognized ("1") or misclassified ("0"). To verify the usefulness of the proposed online optimization method, we compare the model classification accuracy before and after optimization. The results for this experiment are summarized in Table 2. The respective L1 distance results are presented in Appendix B.

## 6 Discussion and Conclusion

Enabled by the complete understanding of the geometry of the spherical classifiers (Banarer et al., 2003b; Perwass et al., 2003; Melnyk et al., 2021), we show in Section 4 that we only need the spherical harmonics of degree up to $N = 1$ to determine the effect of rotation on the activations in 3D. Using this result, we derive a novel 3D steerability constraint (13). The conducted experiment in Section 5.3 shows that the derived constraint is correct since the constructed steerable model produces accurate predictions for the rotated shapes, provided that the rotation is known. From Table 1, we can see that that rotating the data without steering the neurons leads to substantial degradation of accuracy (from 99.2% to 25.2%), whereas using the steerability equation maintains the high accuracy. Moreover, the steerable model classification error only moderately increases with the level of noise in the input data, which is a clear indication of the robustness of the classifier.

The experiment presented in Section 5.4 considers a more realistic setting, where the rotation applied to the input is not known exactly but is inaccurately estimated. However, unlike the first experiment, we perform an optimization to improve the initial steerable model prediction. The only three parameters of the model — the axis–angle representation of the *unknown* external rotation — determining the interpolation coefficients $v^k$, are optimized by minimizing the entropy in the model output. As displayed in Table 2, our steerable model, prior to the optimization, produces rotation-invariant predictions for the transformed shapes for both synthetic and real data, which points to the robustness of our method. The proposed online optimization further improves the model classification accuracy. The lower accuracies for the zero-mean noise (the first row in Table 2) for the skeleton data compared to the $\pi/36$ case (the second row) are presumably caused by a small error in the shape orientations after the performed standardization.

The power of our approach lies in the geometric explainability inherent to the spherical units that constitute our model. Seemingly complicated conformal algebra operations have a surprisingly straightforward interpretation in Euclidean space and allow us to build a steerable feed-forward neural network with a purely geometric motivation. The steerable spherical neurons discussed in our work can be employed in other architectures, e.g., CNNs, and therefore, the equivariance to transformations beyond rotation can be considered.

ETHICS STATEMENT

As the nature of the present paper is mainly theoretical, no direct ethical issues arise, nor are there any immediate societal consequences.

REPRODUCIBILITY STATEMENT

We provide the implementation of the proposed method, as well as notebooks with the conducted experiments, as part of the supplementary material.

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

# A  3D TETRIS DATA

The 3D Tetris dataset (Thomas et al., 2018) is the following set of point coordinates (each shape is of size $4 \times 3$):

```
chiral_shape_1: [(0, 0, 0), (0, 0, 1), (1, 0, 0), (1, 1, 0)],
chiral_shape_2: [(0, 0, 0), (0, 0, 1), (1, 0, 0), (1, -1, 0)],
square:         [(0, 0, 0), (1, 0, 0), (0, 1, 0), (1, 1, 0)],
line:           [(0, 0, 0), (0, 0, 1), (0, 0, 2), (0, 0, 3)],
corner:         [(0, 0, 0), (0, 0, 1), (0, 1, 0), (1, 0, 0)],
L:              [(0, 0, 0), (0, 0, 1), (0, 0, 2), (0, 1, 0)],
T:              [(0, 0, 0), (0, 0, 1), (0, 0, 2), (0, 1, 1)],
zigzag:         [(0, 0, 0), (1, 0, 0), (1, 1, 0), (2, 1, 0)].
```

# B  KNOWN AND ESTIMATED ROTATION EXPERIMENTS: L1 DISTANCES

We compare the hidden unit activations of the ancestor MLGP and constructed steerable models by computing the L1 distance (see Table 3). For convenience, we call the hidden unit activations of the ancestor MLGP fed with the shapes in the canonical orientation *ground truth activations*. From Table 3, we can see that the L1 distance to the ground truth activations linearly and moderately increases with the level of noise in the input data, which is a clear indication of the robustness of the classifier and corresponds well to the model accuracy results in Table 1. Without steerability ("Ancestor" columns), the respective L1 distances increase by one order of magnitude and basically independently of the noise level, demonstrating the significant improvement by the steerability.

Table 3: Known rotation experiment: the L1 distance between the steerable model hidden activations and the ground truth activations given the distorted (the noise units are specified in the square brackets) rotated shapes (mean and std over 1000 runs).

| | 3D Tetris | | | 3D skeleton data (*test* set) | |
|---|---|---|---|---|---|
| Noise $(a)$, [1] | **Steerable** | Ancestor | Noise $(a)$, [m] | **Steerable** | Ancestor |
| 0.00 | **0.00 $\pm$ 0.00** | 8.10 $\pm$ 4.13 | 0.000 | **0.00 $\pm$ 0.00** | 52.58 $\pm$ 30.57 |
| 0.05 | **0.33 $\pm$ 0.05** | 8.07 $\pm$ 4.08 | 0.005 | **0.53 $\pm$ 0.00** | 51.82 $\pm$ 29.36 |
| 0.10 | **0.66 $\pm$ 0.10** | 7.94 $\pm$ 3.99 | 0.010 | **1.06 $\pm$ 0.00** | 50.47 $\pm$ 28.47 |
| 0.20 | **1.32 $\pm$ 0.19** | 8.31 $\pm$ 3.85 | 0.020 | **2.12 $\pm$ 0.01** | 52.69 $\pm$ 29.56 |
| 0.30 | **2.00 $\pm$ 0.31** | 8.26 $\pm$ 3.78 | 0.030 | **3.18 $\pm$ 0.01** | 51.02 $\pm$ 29.44 |
| 0.50 | **3.33 $\pm$ 0.48** | 8.65 $\pm$ 3.44 | 0.050 | **5.30 $\pm$ 0.02** | 51.23 $\pm$ 29.02 |

The results presented in Table 4 indicate that the proposed online optimization, besides improving the classification accuracy (see Table 2), results in the decrease of the L1 distance to the ancestor MLGP softmax output values (referred to as *ground truth output*) for the Tetris data. Note the low absolute level of the error already for the initial result. In case of the 3D skeleton data, the absolute error level is also very low, but we observe a slight increase of the L1 distances after the optimization despite the improved accuracy (see Table 2). We presume it is caused by the increased number of hidden units in the ancestor model compared to the Tetris case (12 vs. 5), which might not all be equally relevant to the classification output.

Table 4: Estimated rotation experiment: the L1 distance between the steerable model softmax output and the ground truth output (mean and std over 1000 runs) for a randomly chosen rotated shape before and after online optimization and different distortion rotation angle $\sim \mathcal{N}(\mu, \sigma = \pi/18)$.

| | 3D Tetris | | 3D skeleton data (*test* set) | |
|---|---|---|---|---|
| Noise angle mean $(\mu)$ | Initial | Final | Initial | Final |
| 0 | 0.007 $\pm$ 0.033 | 0.007 $\pm$ 0.031 | 0.183 $\pm$ 0.331 | 0.196 $\pm$ 0.363 |
| $\pi/36$ | 0.009 $\pm$ 0.038 | 0.007 $\pm$ 0.032 | 0.172 $\pm$ 0.330 | 0.179 $\pm$ 0.353 |
| $\pi/18$ | 0.013 $\pm$ 0.046 | 0.007 $\pm$ 0.044 | 0.231 $\pm$ 0.378 | 0.238 $\pm$ 0.400 |
| $\pi/12$ | 0.025 $\pm$ 0.080 | 0.012 $\pm$ 0.082 | 0.290 $\pm$ 0.409 | 0.293 $\pm$ 0.432 |
| $\pi/6$ | 0.106 $\pm$ 0.205 | 0.064 $\pm$ 0.232 | 0.543 $\pm$ 0.463 | 0.546 $\pm$ 0.482 |

