# OpenReview forum: "Fully Steerable 3D Spherical Neurons"
_ICLR.cc/2022/Conference — ICLR 2022 Submitted_

### Official Review · Reviewer_31zH · 2021-11-02

**Correctness:** 3
**Technical Novelty And Significance:** 3
**Empirical Novelty And Significance:** 2
**Recommendation:** 5
**Confidence:** 2

**Details Of Ethics Concerns:**

N.A.

**Main Review:**

**Strength**
- Writing is clear in general. The relations to prior works are adequately discussed.
- I'm able to follow most of the technical descriptions (although not the details). Some technical terms and notations could be explained better (see minor comments).

**Weakness**
- The motivation of designing rotation-equivariant (or steerable) networks is only briefly mentioned in introduction, and seems rather weak. From reading, one important motivation for steerability is to make rotation-invariant predictions for point cloud classification tasks. However, it is intuitively unclear how to achieve it by steering the pre-trained filters.
- It's not clear how to apply the steeribility constraints to linear layers (MLP). Given that most common architectures are linear, some discussion or explanation on why focusing on spherical neurons is needed.
- The experiment for unknown rotations (Sec 5.4) is weak. Results suggest steerability constraints does not help, where the classification accuracy does not improve significantly after "steering" the weights.
- In Sec 5.4, instead of evaluating classification accuracy, rotation error seems to be a cleaner metric to evaluate the method's ability to recover unknown rotations.

Questions and minor comments:
- The meaning of spherical neuron is not clear. How is it related to hypersphere neuron (Banarer et al., 2003b) and geometric neuron (Melnyk et al., 2021)?
- Given the dense notations, adding a table of notations would help readers keep track.
- The captions and figures are not self-contained. It may help to refer to the corresponding section, or explain some of the notations. Such as in Fig.1, it's not immediately clear what S is.
- Fig. 2: it's not clear which line $t(\theta)$ and $t(0)$ point to
- Fig. 3 caption: what a) and b) refer to is not clear
- In Sec 5.3 known rotation experiments, how are the interpolation coefficients optimized?

**Summary Of The Paper:**

The paper aims to derive a steerability constraint for spherical neurons (3D point classifiers with spherical decision boundary). The steerability constraint enables test-time optimization of a pre-trained classifier to make predictions equivariant to 3D rotation perturbations applied to the input. When input rotation perturbations are unknown, the authors propose a method to recover the unknown rotations and therefore make rotation-invariant predictions. The experiments on a few small scale datasets verifies some of the claims.

**Summary Of The Review:**

Although I carefully reviewed the paper, many of the theoretical claims were beyond my expertise (as indicated by confidence score).

---

> ### Author Response · Authors · 2021-11-12
> **Official Response to Reviewer 31zH**
>
> Thanks for your review and comments.
>
> ### Steerability motivation.
> - Constructing rotation equivariant (steerable) models allows us to use the feature of a given point cloud to synthesize features of the same point cloud with different orientations.
> - Also, rotation equivariant networks enable producing rotation invariant predictions.
>    - In our approach, one can achieve it by computing the interpolation coefficients that, as we show in Eq. 12, directly depend on the orientation of the input, and applying them to the output of the steered pre-trained filters (the spherical filter banks), thereby satisfying the constraint (Eq. 13) - the main result of our paper. (See under “Experiments” for further clarification).
>
> ### Experiments.
> - The experiment in Section 5.3 verifies the derived constraint and checks its stability with respect to noise:
>    - We do not perform any optimization of the interpolation coefficients, but compute them using ground truth input orientations.
> - By the experiment in Section 5.4, we aim to show that the interpolation coefficients $v^k$ (the axis-angle parameters, in that particular case) can be optimized during inference.
>    - The proposed online optimization with the entropy of the model output as the objective is therefore an option we considered worth trying, but it is not the only one.
>    - The results of this experiment are strongly dependent on the optimization formulation rather than on the steerability constraint itself.
>    - The design of more practical online optimization is to be explored in future work.
>
> ### Focusing on spheres.
> - Motivated by the results in the recent work by Melnyk et al. (2021), we adopt their conformal embedding to obtain higher-order decision surfaces.
> In agreement with their motivation of the embedding, we focus on 3D geometry and spherical decision surfaces, arguing for their natural suitability for problems in Euclidean space.
> We base our choice of model construction on the established methodology therein.
> - We explain what we mean by *spherical neurons* in several places, e.g., in Section 1: “a spherical neuron (classifier), i.e., the hypersphere neuron (Banarer et al., 2003b) or its generalization for 3D input point sets — the geometric neuron (Melnyk et al., 2021)”, and the first sentence of Section 3.3: “By spherical classifiers, we collectively refer to hypersphere (Banarer et al., 2003a) and geometric (Melnyk et al., 2021) neurons, which have spherical decision surfaces.” To solidify it further, a spherical neuron = a neuron with a spherical decision surface.
> - The fact that the spherical neurons are steerable is not trivial, nor are the facts that four basis spheres suffice to achieve it in 3D (as we prove in Theorem 4.1.1) and that the tetrahedron construction is a suitable choice for it.
> - The derived steerability constraint is only for the spherical neurons.
> - In general, one can build an ancestor model with the first layer containing spherical neurons, and subsequent linear layers. Following our methodology, one will then build a steerable model by applying the steerability constraint to the first layer (the spherical neurons). As we mentioned in Section 5.2, steerability is not required for the subsequent layers as the output of the first layer becomes rotation-independent.

---

> > ### Comment · Reviewer_31zH · 2021-11-28
> > **Response to rebuttal**
> >
> > Thanks for providing a rebuttal to address my questions.
> > After reading the rebuttal and the comments made by other reviewers, I'd like to maintain my original rating.
> >
> > Overall, the motivation/experiment on steerability still confuses me. In practice, if one would like to make rotation-invariant predictions, they must not know the "ground-truth" rotation that is applied to the input 3D data. Otherwise, one could simply pre-rectify the input 3D data with the ground-truth rotation and make it "in-distribution" to the model (such as a classification network). However, in some experiments, ground-truth rotations are assumed to be known.

---

> > > ### Author Response · Authors · 2021-11-30
> > > **Official Response to the Additional Feedback from Reviewer 31zH**
> > >
> > > Thank you for your additional feedback.
> > >
> > > We agree that requiring ground truth knowledge on the rotation limits the practical use of the method.
> > >
> > > However, the main point of the experiment in Section 5.3 was to show that the ancestor model with rectified input and the steered model produce the same output.

---

> ### Comment · Area_Chair_EVqm · 2021-11-26
> **Please respond to the author rebuttal**
>
> Dear Reviewer 31zH,
> The authors have posted their rebuttal. I wonder whether the rebuttal addressed your concerns? Please respond to the authors. Thanks!
>
> AC

---

### Official Review · Reviewer_CYuQ · 2021-11-02

**Correctness:** 2
**Technical Novelty And Significance:** 3
**Empirical Novelty And Significance:** 2
**Recommendation:** 5
**Confidence:** 3

**Main Review:**


## Strengths.
The submission is tackling a problem of importance for the machine learning community.
The proposed approach is rigorously developed.
Section 5 does show empirical evidence that their approach performs better than the trained MLGP when the input datapoint are rotated.

## Weaknesses.
If my understanding is correct, the model lacks end-to-end training, as it requires a pretrained "ancestor MLGP" model in the first place. What is preventing the steerable layers to be trained directly?

Also, the model seems to require to either know or infer the rotation that has been applied to the input (that the pretrained model saw). If this is indeed the case, I feel that it limits the applicability of the proposed approach quite strongly, as opposed to prior work that solve the constraint for the entire group.

One limitation of the proposed method is the focus on scalar fields, as opposed to some prior work which allow broader classes of feature fields.
One major weakness to me, is the lack of empirical evaluations of baselines (other than the MLGP model from Melnyk et al. (2021)). Why isn't there any comparison with the SO(3)-equivariant CNNs from Weiler et al. (2018a) (and other models cited in Section 2.1), or any point-cloud focused neural networks?

## Clarity.
Overall, the submission is not always easy to follow.
It is perhaps lacking an overview of the method, a sense of where the paper is going. For instance, the Background section feels a bit like a sequence of definitions with no obvious direction or motivation for the reader. I am still not sure to understand why is it useful to work with conformal embeddings. Is it to bypass the design of equivariant non-linearity?

Also, I would argue that framing the steerability condition within the framework given by representation theory & equivariance could help the reader understanding the motivation and the narrative of the submission better.

## Relation to prior work.
As mentioned above, even prior work is indeed discussed properly in Section 2.1, there is no empirical comparison whatsoever with such methods.
As someone that is fairly familiar with the representation theory / equivariance literature, I would be interested to understand how the steerability constraint developed in Section 4 relates with the spherical harmonics basis used in Weiler et al. (2018a)? This would potentially help clarifying the proposed method.

## Additional feedback.
- Title: Why is "fully" necessary? As opposed to partial steerability? In what sense?
- Equations 1 and 2:
    - Is Eq 1 necessary / helping the reader as this paper is focused on the 3D setting?
    - Why not framing this in a representation theory setting? With a linear representation $\rho : SO(3) \rightarrow GL(3)$ such that $\rho(R)$ acts on $f$ according to Eq 1/2.
- Section 3.3:
    - typo "desision"
    - Would perhaps help to define what are "hypersphere and geometric neurons" first?
    - Eq 5: Why is this useful? How does it relate to the distance between $X$ and the hypersphere $S$ (i.e. the distance between $X$ and its orthogonal projection on $S$?
    - Eq 6: Is $N$ the number of points on each point cloud? Cannot this number vary for each cloud? Is there one parametrised spherical neuron for each point? What about the ordering / permutation invariance?
- Section 4.1:
    - Thm 4.1.1: "obtained by rotating S in nD space" -> Would be useful to formally defined what this means, is it for any rotation, or the ones that leave the origin invariant?
    - "rotations $\{R_j\}^4_{j=1}$ must be chosen to satisfy the condition (b) in Theorem 4 (Freeman et al., 1991)." -> Would need to explicit that condition for the reader to understand what section 4.1.2 is about.
    - Eq 9: Shouldn't $R$ acts on $X$ as $R \mapsto R^{-1} X$?
    - Would be useful to write that $f^{R_j}(X) = X^T B(S)$.
- Section 5:
    - Would suggest to move the last paragraph of Section 5.2---which recaps the high level methodology---in Section 4.


**Summary Of The Paper:**

## Summary and contributions.
Authors propose 3D "spherical neurons" leading to rotationally equivariant layers. They do so by building on the spherical and geometric neurons introduced in Melnyk et al. (2021), which leverages the conformal space (for $\mathbb{R}^n$) to perform operations. The authors then solve for the steerability constraint for this neuron and empirically show that the proposed approach overperforms Melnyk et al. (2021) on rotated 3D data.


**Summary Of The Review:**

I find the proposed approach interesting yet its impact seems hindered by some potentially strong limitations, and additionally the empirical assessments do not include relevant prior work.

---

> ### Author Response · Authors · 2021-11-12
> **Official Response to Reviewer CYuQ**
>
> Thanks for your constructive feedback and questions.
>
> ### Motivation.
> Motivated by the results in the recent work by Melnyk et al. (2021), we adopt their conformal embedding to obtain higher-order decision surfaces.
> In agreement with their motivation of the embedding, we focus on 3D geometry and spherical decision surfaces, arguing for their natural suitability for problems in Euclidean space.
> We base our choice of model construction on the established methodology therein.
>
> ### Model construction and relation to prior work.
> - We identify the conceptual difference between our network and the models we refer to in Section 2.1, in particular, the SO(3)-network by Weiler et al. (2018a) as follows:
>     - Instead of constraining the space of learnable parameters, we freely train the ancestor and then make the steerable model with the frozen ancestor parameters.
>    - That is, the related work uses spherical harmonics as atoms, i.e., a hand-designed basis, and learns "only" the linear coefficients under constraints. The only thing we inherit from the hand-design is the constraint of first-degree harmonics (see Theorem 4.1.1); all other DoF are learnable.
> - The steerability requires filters that either are (combinations of) spherical harmonics (e.g., work by Fuchs et al. (2020)) or are learned filters based on spherical classifiers and that behave as such (our work).
> Just learning them blindly could lead to some sort of approximation, but not the exact steerability.
> - To produce invariant predictions, our model does require knowledge of the input orientation (with respect to the canonical pose of the shape that the ancestor has seen).
> It does not limit the applicability of our approach since we show that one, in principle, can optimize them at inference (one possible option is discussed in Section 5.4).
> More efficient ways to perform such optimization are to be further explored.
> - Even though we only consider scalar fields in our work, the result of Theorem 4.1.1 and the derived constraint can be applied for broader classes of feature fields.
>
> ### Experimental comparison.
> - The main focus of our experiment is to verify the derived steerability constraint for spherical neurons and check how stable it is in practice (the noise experiments). Hence, we primarily focus the experimental section on the comparison with the ancestor itself (MLGP), rather than on benchmarking with other point-cloud processing networks.
>  - At the end of Section 5.3 (the last sentence), we did mention the results of the analogous Tetris experiments conducted in prior work.
> Initially, we even appended that sentence with an explanation of why this comparison is not quite fair due to the conceptual difference between the methods (as we pointed out above).
> We have now added it back to the manuscript.
> - Thanks to your comment, we have identified yet another difference between the methods (and experimental setups): Weiler et al. (2018a) represent the input data by a specifically chosen 3D grid and Thomas et al. (2018) expand the input values to an exhaustive binary relation (differences of all points).
> - Also, neither Thomas et al. (2018) nor Weiler et al. (2018a) considered the case of perturbed input shapes. The effect of input perturbations is especially interesting in the latter case since the method operates on regular grids.
>
> ### Additional comments.
> - “Fully” was meant to emphasize the rigidity and explicitness of the derived steerability constraint (Eq. 13). We can change the title to "Exactly Steerable 3D Spherical Neurons" if you suggest it can improve our manuscript.
> - Section 3.3:
>    - Eq. 5 is used to construct a spherical decision surface, i.e., the hypersphere neuron, and implement it with the standard scalar product  (Banarer et al. (2003a,b), Melnyk et al. (2021)) ;
>   - Eq. 6 is the definition of the geometric neuron from Melnyk et al. (2021). Therein, $N$ is the number of points in the input point cloud, which, in general, can vary, but not in the classification tasks considered in our and prior works. This means, for each input point, a geometric neuron learns a respective sphere. Permutation invariance is not attained with the baseline model. However, we derive the steerability constraint for a single spherical decision surface a.k.a the hypersphere neuron (sphere) taking one point as input (see Eq. 9). One can address the problem of permutation invariance quite straightforwardly by following the construction of, e.g., PointNet, replacing the MLPs applied to each point in isolation with spherical neurons and then applying a global aggregation function to the extracted features.
> - Section 4.1:
>    - Theorem 4.1.1: rotations in $nD$ space mean that we consider $SO(n)$, i.e., leaving the origin invariant.
>    - Eq. 9: in our work, a rotation $R$ acts on a point $X$ as a left-hand-side operator, i.e., as $RX$.
>    - $f^{R_j}(X) = X^⊤ R_j S$. (The input $X$ is a $5\times1$ matrix and $B(S)$ is a $20\times1$ matrix.)

---

> > ### Comment · Reviewer_CYuQ · 2021-11-28
> > **Re: Official Response to Reviewer CYuQ**
> >
> > I thank the authors for taking the time to reply to my questions and remarks.
> >
> > I may have wrongly classified some key features of the proposed approach as weaknesses. These include a two-stages training (first ancestor model and then the DoF from the steerability constraint) and the need to infer the input's rotation. As these features differ from previous methods it would be particularly interesting to see whether these choices are beneficial. Yet, the authors explicitly state that a `comparison is not quite fair due to the conceptual difference between the methods` which I find a bit startling.
> >
> > I find the proposed approach interesting and sound but I feel that the exposition is still a bit convoluted and that the empirical evaluations are too limited as per my aforementioned remark.
> > Henceforth, I stand by my original score.

---

> > > ### Author Response · Authors · 2021-11-29
> > > **Official Response to the Additional Feedback from Reviewer CYuQ**
> > >
> > > Thank you for your additional feedback.
> > >
> > > We are sorry if our rebuttal was unclear about the experimental comparison.
> > >
> > > What we meant by our comment about the comparison of the Tetris experiment results (Section 5.3) of our and the related methods (by Thomas et al. (2018) and Weiler et al. (2018a)) was to address your original remark that `there is no empirical comparison whatsoever with such methods.`
> > >
> > > As we stated in our first response, at the end of Section 5.3, we did mention the empirical results of prior work.

---

> ### Comment · Area_Chair_EVqm · 2021-11-26
> **Please respond to the author rebuttal**
>
> Dear Reviewer CYuQ,
> The authors have posted their rebuttal. I wonder whether the rebuttal addressed your concerns? Please respond to the authors. Thanks!
>
> AC

---

### Official Review · Reviewer_FkmY · 2021-11-03

**Correctness:** 3
**Technical Novelty And Significance:** 3
**Empirical Novelty And Significance:** 3
**Recommendation:** 8
**Confidence:** 4

**Main Review:**

This paper relies heavily on the technical content of the Melnyk 2021 paper. The key idea in that paper is to us a type of conformal embedding (as reviewed in Section 3.2) whereby non-linear Euclidean distance is transformed to a simple scalar product. This space is of dimension R^n+2, where n is the dimension of the original space. This construction is at the heart of the geometric neuron (illustrated in Fig. 1). The approach taken in the present paper is to build a fully steerable version of this geometric neuron.

Strengths: The technical contributions of the paper are clear and are largely associated with Sections 4 and 5 (building on the past work and the Melnyk 2021 paper, reviewed in Sections 2 and 3). The paper shows how many basis functions are needed (a total of 4 in three dimensions) and then provides a construction using a tetrahedron to place these four basis functions. This results in a 4 x 5 = 20 x 1 matrix representing the filter bank of the classifier (eq. 8). Using geometric reasoning, the steerability constraint is then derived in explicit form (eq 13). [Note: I did not check all the math, but to me the reasoning and reliance on past work seems sound.]

Weaknesses: The present experiments demonstrate a type of proof of concept. This is fine in my view. The actual potential of steerable spherical neurons could be better shown in future work, particularly given the relevance of the contribution to ICLR in terms of new ideas for "representation" learning.

**Summary Of The Paper:**

The paper proposes a method for constructing steerable spherical neurons, building on the recent Geometric Neurons developed in Melnyk et al 2021. The main technical result in the paper is a steerability constraint for a geometric neuron, as given by eq. (13) in the paper. This constraint is used in an implementation whereby a steerable model is constructed and is then use for two tasks. The first task involves the use of steerable spherical neutrons for the classification of 3D Tetris objects seen under rotations, and the second is a similar experiment applied to 3D skeleton data. The results demonstrate a very large performance boost when using the steerable versions. The second experiment builds on this idea to construct a version that adapts a possibly imperfect initial rotation estimate, using the representation. This second experiment is in the spirit of demonstrating equivariance under 3D rotations with perturbations.

**Summary Of The Review:**

Whereas this paper does rely on some heavy lifting in the recent Melnyk 2021 paper (which itself builds on past work), this past work is carefully reviewed in Sections 2 and 3. The meat of the novel contribution is in Section 4. The derived steerability constraints in the present paper and the construction of the basis spheres using placement on vertices of a tetrahedron, followed by the proof of concept use of the steerable spherical neurons, for particular tasks (e.g. recognition from skeletons seen under rotation), demonstrate the advantages of the approach. Whereas this might by some be considered by some to be a niche contribution, and others might argue that the experimental results could be strengthened, I for one appreciate the geometric reasoning and the relevance to ICLR themes. Overall the paper is quite dense and in places there are sentences that I was not able to parse, e.g., "Equivariance is a necessary property for steerability as the group acting on the input space needs to be represented in the co-domain of the operator". However, I consider these issues to be somewhat minor. I believe the strengths sufficiently outweigh any such weaknesses.

In terms of demonstrated applications of 3D steerable geometric neurons, I think that could come later. This is not primarily pitches as a benchmark oriented or empirical results derived paper. Rather, it is an attempt to add new and likely quite useful theory, and methods to apply that theory.

---

> ### Author Response · Authors · 2021-11-12
> **Official Response to Reviewer FkmY**
>
> Thanks for your constructive feedback and comments.
>
> We highly appreciate the depth of your review and understanding of the main idea we try to convey through our work.
>
> By the sentence in Section 2.1 you mentioned, we mean:
> - Steerability requires changing the output depending on the actions of a certain group.
> - Thus, the output needs to contain a representation of the group that acts on the input.
> - Therefore, the operator must commute with the group (note that the representation might change from input to output).

---

> ### Comment · Area_Chair_EVqm · 2021-11-26
> **Please post your post-rebuttal opinion**
>
> Dear Reviewer FkmY,
> The authors have posted their rebuttal. I wonder whether the rebuttal addressed your concerns or whether other reviewers' comments changed your evaluation? Please respond. Thanks!
>
> AC

---

> ### Comment · Reviewer_FkmY · 2021-11-29
> **Response to authors and reviewers**
>
> First, I'd like to thank the authors for their response and the other reviewers for their active discussion. I have considered the critique that the other reviewers have provided, which primarily has to do with a lack of evaluation on benchmarks. The other reviewers also comment in part on the need for better exposition.
>
> For a conference like ICLR I for one would welcome new ideas, and steerability in 3D in a neural model is new. Having worked in areas related to vision for over two decades and having seen 100 papers this year alone as an area chair in vision meetings, I am struck by how benchmark oriented our community has become. This is stifling good ideas, and in a sense progress.
>
> I realize that my opinion is at odds with that of the other reviewers, but I feel that the authors have presented their ideas clearly enough, with a well motivated algorithm, and although I concur that the experiments are proof of concept (as stated in my original review) I'm willing to cut them some slack on that front. I think that full scale  empirical evaluation is not necessarily a condition for acceptance. Hence, I would like to stand by my original positive rating of this paper.

---

### Official Review · Reviewer_KgA9 · 2021-11-08

**Correctness:** 3
**Technical Novelty And Significance:** 3
**Empirical Novelty And Significance:** 2
**Recommendation:** 5
**Confidence:** 2

**Main Review:**

Two main strengths of this paper are as follows:
1, The method proposed in the paper is very novel.
2. The figures are well drawn for clearer illustration.
3. The contribution of the paper is list very clear.

Besides, there are some weaknesses of this paper:
1, The relation between the model and MLP need be further discussed, in addition, advantages compared to other equivariant methods should be stated.
2, The baseline of experiments is lacking, and more methods on point cloud classification need to be compared.

--------------------------------Post rebuttal----------------------------------

Thank you for your rebuttal. I agree with other reviewer that experiments of comparison with other equivariant model on more challenging and mearningful tasks should be provided. Althoud I am quite familiar with topic of equivariance, I have to admit I have a lot of difficulty undertanding the paper, that's why I score a very low confidence in the initial review. After reading other reviewer's comments, I give my additional comments. There are some unclear points about the theoretical part of the paper.

 In the last line of section 2.1, it is claimed that steerable model is constructed from freely trained base network, while, in the (13), the paper presents the main results, the steer constraint, which seems to be contradictable to me. Could the author give more explanation, what is the 'steerable' means here and what is the main difference with the previous steerable model.  In addition, from my perspective, steerable constraint shoud be followed by a method to solve it, while I can not find it in the paper(please correct me if I miss something).

About the experiments, I want to know whether the model is exact equivariant? As it is mentioned in the 3rd paragraph in the section5.2, the output of the first layer is rotation-independent, how could a rotation-independent model be used for rotation prediction？

**Summary Of The Paper:**

This paper introduces steerable 3D neurons which are equivariant to 3D rotations for point cloud classification. Theories have been provided to support the steerability of the model, and experiments are carried out on synthetic and real 3D data to verify the equivariance proved in theories and effectiveness of the model.

**Summary Of The Review:**

While the paper has much theoretical parts, experiments on more complicated dataset and comparisons to recent equivariant methods are needed to verify the soundness of the method.

---

> ### Author Response · Authors · 2021-11-12
> **Official Response to Reviewer KgA9**
>
> Thanks for your review and comments.
>
> - Our steerable model construction is based on the ancestor MLGP by Melnyk et al. (2021), who describe in detail the relation to the standard MLP.
>
> - The main focus of our experiments is to verify the derived steerability constraint for spherical neurons (Eq. 13 - the main result of our work) and check how stable it is in practice (the noise experiments).
>
> - Hence, we primarily focus on the comparison with the ancestor itself (MLGP) rather than on benchmarking with other point cloud processing networks.
>
> - Constructing larger baseline architectures with the considered spherical neurons for performing the suggested comparisons with other point cloud classification methods is to be done in future work.

---

> ### Author Response · Authors · 2021-11-15
> **Official Response to the Additional Feedback from Reviewer KgA9**
>
> Thank you for your additional feedback.
>
> We are happy to clarify the unclear points.
>
> ### Steerable model construction.
> - Our steerable model is obtained by freezing the trained ancestor--base network--parameters and transforming its first layer into spherical filter banks,  Eq. (8).
> - The responses of the filter banks {$B(S)$} (i.e., the first layer of the steerable model) will be interpolated by the coefficients {$v^k$} (Eq. (12)) that are the only free parameters of our steerable model.
> - "Steerable" means that an arbitrarily rotated copy of the ancestor network is obtained by means of a linear combination of four fixed rotated copies. The underlying spherical neurons (forming the filter bank) are combined with linear coefficients that are determined to fulfill (13).
> - The above is shown by Eq. (9) -- the steerability constraint for a single sphere (i.e., a hypersphere neuron).
> - Could you clarify what is meant by “the previous steerable model”?
> - Eq. (13)--the steerability constraint for a geometric neuron--is used only for constructing the steerable model and does **not** concern the training of the ancestor (base) network that we train unconstrained, just as Melnyk et al. (2021).
>
> ### Steerable model training and experiments.
> - We are uncertain what you mean by the qualified form “exact equivariant”, in contrast to just “equivariant”. Could you clarify what you mean by the model being “exact equivariant”?
> - To be clear, the first layer of our steerable model (the output of the filter banks before the interpolation) is equivariant to rotations.
> After we have determined the interpolation coefficients  $v^k$, either by a known rotation or optimization, and applied them to the filter banks output, the output becomes invariant, and so does the output of the entire network.
> - *“...steerable constraint should be followed by a method to solve it”*:
>     - To solve the constraint is to compute the interpolation coefficients (Eq. 12).
>     - By the experiment in Section 5.4, we aim to show that the interpolation coefficients $v^k$ can be optimized during inference. The proposed online optimization with the entropy of the model output as the objective is therefore one option we considered worth trying, but it is not the only one. The design of more practical online optimization is to be explored in future work,
>
> ### Rotation prediction.
>  - Since, as we show in Eq. (12), the coefficients $v^k$, directly depend on the orientation of the input, we can optimize with respect to the unknown rotation parameters, and thus interpret the solution as the rotation prediction.

---

> > ### Comment · Reviewer_KgA9 · 2021-11-26
> > **Response to rebuttal**
> >
> > Thanks for your detailed rebuttal for addressing my concern. However, as experiments for comparison with other equivariant model are still not provided, I remain my original score.

---

> ### Comment · Area_Chair_EVqm · 2021-11-26
> **Do you have further comments?**
>
> Dear Reviewer KgA9,
> The authors have posted their rebuttal again. I wonder whether the rebuttal addressed your concerns? Please post your further opinions. Thanks!
>
> AC

---

### Official Review · Reviewer_uUFZ · 2021-11-08

**Correctness:** 3
**Technical Novelty And Significance:** 3
**Empirical Novelty And Significance:** 2
**Recommendation:** 5
**Confidence:** 2

**Details Of Ethics Concerns:**

The paper is of theoretical nature. I do not see any immediate ethical issues.

**Main Review:**

PROS
- I find the steerability addition to the existing geometric neurons to be valuable and to a certain degree novel.
- The paper is well-structured, despite some issues with clarity. The writing style makes it understandable.
- I did not spot theoretical issues. I believe that the approach is technically sound.
- The experiments are sensible albeit simplistic.

CONS
- The contributions of this paper are not written in a concise manner and clutters the main message of the paper. I would suggest revising them for the sake of clarity and understandability. e.g. the number of contributions can definitely be reduced.

- Introduction does not well motivate what applications would be enabled if the posed problem were to be solved. If the steerability were to be introduced into the geometric neurons, what benefit do we get? The same also holds for limitations of the approach. I do not see them to be discussed. For instance, does the two-stage training approach pose any issues in applications?

- In general, I believe that the writing style of this paper is a bit convoluted. For example, it is important to clearly state from the beginning that the paper proposes an inference-time add-on onto neural networks composed of geometric neurons. And how are the 'online optimization' and the 'learnable parameters' related? As far as I understand (13) is optimized for v^k during test-time to obtain rotation-invariant classification? If this is not right, I might have a gap in my understanding. (Please also define what is truly meant by a 'steerable model'.) In fact, is it better to have a separate test set for optimizing {v^k}?

- Could section 4.1.2 or any other preceding section explain why spherical filter banks are necessary and how they would be used? - before explaining 'how' they are constructed? This might make it easier for people who are not strongly versed in signal processing, i.e. the connection with basis functions and some filter banks could be made clearer.

- The method section can end with a summary of how Eq. 13 is used. - or if this is to be relegated to experimentation, what follows after Eq. 13 could say so.

- Can the paper speak more about why a hypersphere can be seen as a generalization of a hyperplane in the context of the paper? (i.e. in general one cannot recover the plane from a sphere.)

- In Sec. 5.3, some noise seems to be added to the point coordinates. I am wondering if this is added in addition to the random rotation or applied separately.

- Why call Sec. 5.4 'Adaptive' ? This seems like a prior/condition rather or simply an initial estimate, which is 'refined'.

- The entropy loss used in Sec. 5.4 seems to be a proxy to train the 'steerable model'. First, I am a bit surprised that this is even sufficient. Why not apply the same idea to many of the CNNs then? And this intuition seems to be confirmed by Table 2 - the initialization is as good, and almost no benefit is brought by the subsequent step (online optimization). Second, instead of choosing just the entropy, why not have a proxy rotation loss? Even at inference time, one could transform the object multiple times. In fact, the interplay between rotation and classification losses inspired other equivariant networks:
* Zhao, Yongheng, et al. "Quaternion equivariant capsule networks for 3d point clouds." European Conference on Computer Vision. Springer, Cham, 2020.

A paper not be missed in this context. Also note that such rotation invariant classifiers have a common evaluation protocol (e.g. using ModelNet/ShapeNet) as explained in that paper (Zhao et al.). I would strongly suggest to compare against some of these methods. In fact, at the moment the paper lacks any sort of a comparison to the state of the art except the 'ancestor' baseline, which is not rotation invariant anyway.

- I notice that this approach can estimate the correct pose of the object with some theoretical guarantees (not discussed in the paper). I would advise that the paper considers the common SE(3)/SO(3) pose estimation benchmarks based on ModelNet/ShapeNet datasets. This way a plethora of comparisons can be made.

- The paper mentions 'geometric explainability'. This notion is in fact interesting and explainable networks are desirable. Can we see an evaluation/study where this is fulfilled?

Minor Notes:
- Since we are at inference -> Since at inference

**Summary Of The Paper:**

This paper proposes to make the geometric neurons of Melnyk et al.'21 to be steerable so that objects undergoing arbitrary rotations can be classified with higher accuracy. This is done in multiple stages. First, the neurons of Melnyk et al. are trained to convergence with a hyperspherical output layer. Then, the frozen weights are transformed such that the input of a *steerable neuron* can be written as a linear combinations of the rotated versions of itself. The experiments act as a sanity check while demonstrating the validity of the algorithm on a simple human pose dataset.

**Summary Of The Review:**

With more transparency, clarity and experimental evaluations, this paper would be very strong.

---

> ### Author Response · Authors · 2021-11-12
> **Official Response to Reviewer uUFZ**
>
> Thanks for your constructive feedback and suggestions.
>
> Below we address your comments.
>
> ### Related work.
> - Thank you for recommending the paper by Zhao et al. (2020). We have now included a reference to it in the manuscript.
> - The fact that a hypersphere is a generalization of a hyperplane, i.e., hyperplane = hypersphere with infinite radius, is pretty standard in the considered embeddings. It mainly concerns the prior work by Banarer et al. (2003a,b), Perwass et al. (2003), and Melnyk et al. (2021).
>
> ### Steerable model.
> - Our steerable model is obtained by freezing the trained ancestor parameters and transforming its first layer into spherical filter banks, the responses of which will be interpolated by the coefficients $v^k$ that, as we show in Eq. 12, directly depend on the orientation of the input and are the only free parameters of our steerable model. We agree that calling them "learnable" is ambiguous. Therefore, we have changed it in the manuscript to "free".
> - Your understanding is fully correct: Online optimization, i.e., during inference, refers to optimizing these $v^k$ to obtain rotation-invariant classification predictions.
> - But since the $v^k$ depend on the input rotation, we can optimize with respect to, e.g., the axis-angle representation parameters of this rotation (as we do in the experiment in Section 5.4).
>
> ### Experiments.
> - In Section 5.3, noise is added to point coordinates after they have been transformed with a random rotation.
> - We considered “Estimated” as an alternative name for the experiment in Section 5.4. It may indeed be a better choice, so we changed it in the manuscript.
> - By the experiment in Section 5.4, we aim to show that the interpolation coefficients $v^k$ (the axis-angle parameters, in that particular case) can be optimized during inference. The proposed online optimization with the entropy of the model output as the objective is therefore one option we considered worth trying, but it is not the only one.
> - To test this online optimization, one can use the same test split, which by definition was not used to either optimize the ancestor parameters or determine the training hyperparameters.
> - The design of more practical online optimization is to be explored in future work, as well as constructing larger baseline architectures with the considered spherical neurons for benchmarking on ModelNet/ShapeNet and performing the suggested comparisons.
>
> ### Addressing other comments.
> - We have now revised the contribution list (see the end of Section 1 on p.1).
> - To make a clear connection between basis functions and spherical filter banks, we start Section 4.1.1 by saying “To formulate a steerability constraint for a spherical neuron (sphere), first, we need to determine the minimum number of basis functions…” and the third paragraph in Section 4.1.2 reads “the four rotated versions of the function $\textit{f(X)}$ will constitute the basis functions that we call a spherical filter bank.”
> - The “geometric explainability” of our approach is two-folded:
>    - It stems from the use of 3D spherical decision surfaces, which provide us with an interpretable decision-making process, as described in detail in Melnyk et al. (2021) (especially Section 4 and Figures 2 and 4), who argue for their natural suitability for problems in Euclidean space;
>    - The tetrahedron construction of the filter banks is intuitive and easy to visualize.

---

> ### Comment · Area_Chair_EVqm · 2021-11-26
> **Please respond to the author rebuttal**
>
> Dear Reviewer uUFZ,
> The authors have posted their rebuttal. I wonder whether the rebuttal addressed your concerns? Please respond to the authors. Thanks!
>
> AC

---

### Comment · Area_Chair_EVqm · 2021-11-28
**Please post your post-rebuttal opinion!**

Dear Reviewers,
The authors have updated their manuscript and responded to your comments. Please check whether your concerns have been addressed and then post your further opinions *if you haven't*. This is the professional way to show respect to the authors' efforts. The deadline Nov. 29 is coming very soon. Thanks!

AC

---

### Decision · Program_Chairs · 2022-01-20

**Decision:**

Reject

**Comment:**

The paper proposed a new kind of neurons for 3D spherical data classification. All the reviewers agreed that the new kind of neurons makes a good contribution. However, all the reviewers also agreed that the experiments are too weak: only at the proof-of-concept level and no comparison with the state of the arts. Only reviewer FkmY advocated accepting the paper because we need new ideas, and all other reviewers leaned towards rejecting the paper. The AC had exactly the same feeling as the reviewers. Particularly, the AC also agreed with reviewer FkmY that we should not look at experimental results only. However, the AC would like to point out that this by no means means that the experiments can be too simple. Note that this paper is to propose a new tool to improve classification performance, rather than a new theory to explain or predict something. So some basic requirements on the experiments are necessary. If the authors could provide comparison with the state of the arts and with reasonably good performance, not necessarily exceeding or even on par with the state of the arts (namely can be inferior but not too inferior so that others can believe adding engineering tricks could fill in the gap), the AC would consider accepting the paper.